# Smartphone Apps for Domestic Violence Prevention: A Systematic Review

**DOI:** 10.3390/ijerph20075246

**Published:** 2023-03-23

**Authors:** Mehreen Sumra, Sohail Asghar, Khalid S. Khan, Juan M. Fernández-Luna, Juan F. Huete, Aurora Bueno-Cavanillas

**Affiliations:** 1Department of Computer Science, COMSATS University Islamabad, Islamabad 45550, Pakistan; 2Department of Preventive Medicine and Public Health, University of Granada, 18071 Granada, Spain; 3Department of Computer Science and Artificial Intelligence, University of Granada, 18071 Granada, Spain

**Keywords:** domestic violence, violence prevention, smartphone apps, systematic review, intimate partner violence, violence against women

## Abstract

Smartphone applications or apps are increasingly being produced to help with protection against the risk of domestic violence. There is a need to formally evaluate their features. Objective: This study systematically reviewed app-based interventions for domestic violence prevention, which will be helpful for app developers. Methods: We overviewed all apps concerning domestic violence awareness and prevention without language restrictions, collating information about features and limitations. We conducted searches in Google, the Google Play Store, and the App Store (iOS) covering a 10-year time period (2012–2022). We collected data related to the apps from the developers’ descriptions, peer reviewed research articles, critical reviews in blogs, news articles, and other online sources. Results: The search identified 621 potentially relevant apps of which 136 were selected for review. There were five app categories: emergency assistance (*n* = 61, 44.9%), avoidance (*n* = 29, 21.3%), informative (*n* = 29, 21.3%), legal information (*n* = 10, 7.4%), and self-assessment (*n* = 7, 5.1%). Over half the apps (*n* = 97, 71%) were released in 2020–22. Around a half were from north-east America (*n* = 63, 46.3%). Where emergency alerts existed, they required triggering by the potential victim. There was no automation. Content analysis showed 20 apps with unique features, including geo-fences, accelerometer-based alert, shake-based alert, functionality under low resources, alert auto-cancellation, anonymous communication, and data encryption. None of the apps deployed artificial intelligence to assist the potential victims. Conclusions: Apps currently have many limitations. Future apps should focus on automation, making better use of artificial intelligence deploying multimedia (voice, video, image capture, text and sentiment analysis), speech recognition, and pitch detection to aid in live analysis of the situation and for accurately generating emergency alerts.

## 1. Introduction

The World Health Organization (WHO) defines violence as the “intentional use of vulnerable or actual physical force or power against oneself, another person, a group, or a community that results in or has a high probability of resulting in injury, death, spiritual harm, maldevelopment, or deprivation” [1]. Domestic violence is defined as any act of gender-based violence that can cause physical, sexual, or mental harm or suffering to women, including threats of such acts, coercion, or arbitrary deprivation of liberty, whether committed in public or in private life [2]. The condition of domestic violence suffered by women is widely prevalent [3]. Cell phone apps have the potential to be used for security and defense, for example, women may contact nearby family, friends, and organizations for assistance in conflict or dangerous situations. This approach was more successful than using pepper spray [4].

Domestic violence is one of those conditions that may benefit from use of apps. In this era, mobile phones are obligatory gadgets, as they provide the perfect way to get connected with others, with 6.567 billion people using smartphones [5], which is 82% of the global population [6]. According to real-time intelligence data, there are more than 10.57 billion mobile connections globally, exceeding UN digital analyst estimates of 7.93 billion [7]. The communications between a mobile device and a cell tower are referred to as mobile connections [8], and there are 2.64 billion more mobile connections than humans on the earth [7]. Smartphones provide multiple benefits, such as helping to meet personal, entertainment, and business needs, but they can also support crime prevention by means of pieces of software called applications or apps. In the first quarter of 2021, 3.48 million and 2.22 million apps were available on the Google Play Store and Apple App store, respectively [9].

In healthcare, mobile-based apps are already in use to educate and to manage diseases, for example, diabetes treatment [10], weight management [11], smoking prevention [12,13,14], suicide avoidance [15], and human immunodeficiency virus [16,17,18,19,20]. An analysis was performed that identified only 38 dating and domestic violence apps from only one platform (iPhone) using the K-means cluster [21]. Another analysis of online interventions was done, and the authors found that the majority of interventions focused on personal safety plans that included ending the relationship with the partner [22]. A systematic review was conducted, in which the authors classified the domestic violence prevention apps by structured qualitative content analysis [23]. The review relied on a search of app stores (iTunes and the Google Play Store) to identify relevant apps, which may have missed some relevant apps that are not available on those app stores or that were not included in the search criteria. The review was directed to frontline health and social workers [23]; it did not detail the features of the apps. Another previous review [24] assessed the quality of applications related to sexual violence and intimate partner violence using the Mobile Application Rating Scale (MARS), a tool to evaluate apps on a variety of health-related issues; it also did not describe the particular features of existing apps. In another paper, the authors described the design and development of a suite of web-based apps for screening and safety planning related to intimate partner violence (IPV) [25], resulting in a list of user-friendly apps that were customizable to the needs of different service providers and populations [25]. In our study, we addressed the distinguishing characteristics of the apps and provided a critical analysis.

Many apps are available online, which can send a Short Message Service (SMS) text to pre-selected numbers with Global Positioning System (GPS) location by pressing some sort of alarm or alert buttons. Some apps have features that connect users to the nearest police station, a specific country’s hotline number, non-governmental organizations (NGOs), shelter homes, and hospitals [26]. Some apps function uniquely, as they include a questionnaire and a guide to help reduce the amount of violence. The main goal of these apps was to raise awareness about violence and victims’ rights. The victim may also be connected to support services such as lawyers, shelters, social workers, and hospitals through these apps. One of the most significant problems that victims face was the need to provide evidence once the incident has been reported. Voice recording, image capture, and video recording are available through mobile phone apps for this purpose. According to a recent search in Google Scholar combining the terms “domestic violence apps” with “systematic review” and supplementing with searches of IEEE Xplore, ACM Digital Library, and Springer (April 2022), there had been no systematic data compilations.

Given the above background, we undertook a systematic review of app-based interventions and related research on domestic violence prevention.

## 2. Methods

We followed robust systematic review methods and applied the MOOSE reporting guidelines [27].

### 2.1. Search

An online search strategy was used to identify apps that deal with domestic violence, particularly those that deter violence against women. In addition to “domestic violence”, the following keywords and related terms can be found in library databases: women’s security, gender-based violence, intimate partner violence, family violence, wife abuse, battered women, partner abuse, violence against women, domestic abuse, and interpersonal violence [28]. We carried out a systematic online search on Google and two app platforms: the Google Play Store (Android) and the App Store (iOS). In order to find all apps related to domestic violence, we searched all the above terms. The search queries were launched between 1 September and 13 September 2022, and they covered results from 2012 to 2022. We conducted the search of last ten years because the apps in this decade covers all the features of previous apps. We have selected some regions for our research eligibility criteria because we found most the apps on domestic violence from these regions. Communication, discussion, and consensus were used to resolve disagreements about the apps’ eligibility.

### 2.2. Screening

Apps related to domestic violence are described in the identification section of Figure 1. Screening was applied to the title and description of the app on the app platforms. Some apps were discovered through a Google search and were not available on both platforms. However, duplicates were removed and these apps were counted only once.

### 2.3. Eligibility

Apps fulfilling all the following inclusion criteria were included in our systematic review:Apps released or updated after 2011 and till September 2022.The number of downloads of the apps > 100.The language of the apps English, Spanish, Urdu, or multilingual (English and local languages of the region).Applications were chosen from certain geographical regions based on the limitations placed by the download availability of apps, the lack of apps available with distinctive features and duplication of apps.

### 2.4. Data Extraction and Synthesis

App descriptions and critical reviews on platforms, respective app websites, news articles, and other online sources were used to extract data for eligible apps. The following variables were coded using the information gathered: application name, country, launch date, features, distinctiveness, and critical reviews—including positive and negative user experiences—from consumer platforms. After classifying the 136 selected apps into five functional categories (see Figure 2a), content analysis was applied to shortlist the apps with unique features for a detailed description.

## 3. Results and Discussion

Our online search revealed 621 apps. By eliminating 253 duplicates, 368 apps remained for the app title and descriptions screening. After the elimination of 232 apps that did not fulfil the selection criteria, 136 apps were selected for systematic review. After the delineation of their features, finally 20 apps were eligible for detailed evaluation (see Figure 1).

### 3.1. Classification of the Apps

The 136 apps were developed to ensure the prevention of domestic violence in the following classification (see Figure 2 and Appendix A):Emergency assistance apps was the first category that included the majority of the apps. These apps were created to keep individual users safe. Most of them worked in such a way that they asked for emergency contact numbers during registration, and the app can call these numbers in an emergency. Users can trigger Save Our Souls (SOS) alerts. Some apps allowed users to seek assistance by sharing their current location and sending an alert message to pre-selected numbers.Avoidance apps fell into the second category that provides victims with a strategy for avoiding domestic violence situations, as well as emergency assistance.Informative apps fell into the third category that educated users on how to recognize and respond to violent situations.Legal information apps was the fourth category that includes all the apps that provide legal information or terms related to the law. These apps assist users in understanding legal terms by providing books of law and violent acts. Victims can also contact police, law enforcement agencies, firefighters, and hospitals by pressing a designated button on these apps.Self-assessment apps was the final category, and these apps assist users in understanding the term violence and determining the possibility of danger or abuse in their relationship. Through these apps, users may be able to connect with services such as lawyers, shelter homes, social workers, NGOs, and hospitals.

### 3.2. Trends with Regard to Apps Distribution

Emergency apps (see Figure 2a) are the majority of domestic violence apps (45%) worldwide. Figure 2a shows that avoidance apps and informative apps (21%) were the second most popular, followed by legal information apps (8%), and self-assessment apps (5%). Emergency apps were the most popular functional category in many regions (see Figure 2b). A quarter of the apps were released from Southeast Asia (*n* = 35, 25.7%) and around a half were from Northeast America (*n* = 63, 46.3%).

Figure 2c indicates that emergency apps were the most downloaded ones. Under a fifth of the apps (*n* = 28, 20.6%) were downloaded >50,000 times. Therefore, emergency apps were in high demand. As the majority of apps were free to download, everyone has access to a variety of app-based interventions. Many apps that include call or SMS sending functions, such as emergency apps, come with associated charges for such calls, which were imposed by the customer’s network provider, not by the app (via download or in-app purchases). As a result, access and use of many apps with that type of restriction was difficult, and that is why they were not considered for this review. Figure 2d shows that when comparing the years 2012 and 2022, app development rates increased by a factor of eleven-and-a-half. There were also changes in the main function distribution of developed apps, with a noticeable relative increase in the development of informative, self-assessment, and legal information and law-related terms apps (see Figure 2d).

Emergency apps for domestic violence comprise the largest proportion of apps by type of app function, as shown in this review. This trend was particularly strong in certain regions with a high rate of domestic violence in north-east America and South Asia (see Figure 2b). Moreover, the focus of the development of apps should move towards more awareness apps. This way, the incidents could be minimized before occurrence. These graphs clearly shows that these apps have grown in popularity over the last half decade and people are using mobile apps for domestic violence prevention.

Table 1 listed out the features and other details of the various types of above-mentioned apps to stop and prevent violence. These reviews will be helpful to improve the functionality of the potential app.

### 3.3. Critical Analysis

The applications, according to Table 1, serve as a central point of contact for victims and rescue resources. It applies to emergencies only by generating an alarm. Alarm generation is very simple by clicking one alarm button, shaking, or screaming. Some apps help the users by guiding them on how to avoid domestic violence incidents before they happen. When going out alone, some apps, such as StaySafe, BSafe, and My Angel Guard, have a feature called “Follow Me” or “Watch over Me,” which allows the user to share their location and status with contacts. These apps are also used to combat street violence or crime.

ShelterSafe is an online database having interactive map used to connect women with the nearest shelter that can provide them with protection, courage, and assistance [42]. A map serves as both a tool and a reminder of the safety net that has been established across Canada to assist victims of domestic violence. It allows the victims to hide the website by redirecting to another website, which encourages the victim to seek help by preventing the abuser from seeing her [42,43].

Women were hesitant to use the app for several reasons. One aspect was that the apps have a registration form, which discourage people from using it. Users expect easy accessibility, as other apps enable them to sign up using their Facebook or email accounts. Second, users want to be able to contact emergency services for free. However, some apps, such as BSafe and StaySafe, charge a lifetime fee, a monthly subscription fee, and may charge for premium features. The “Follow Me” or alert button, which sent an alert to pre-selected emergency contacts or the country’s emergency service, were examples of premium features. These apps were for people who can afford to buy a smartphone and pay for an internet or data connection, both of which were more expensive than traditional mobile phones. If the victim is completely reliant on the perpetrator and pays the app fee with a credit card, this option may be more dangerous, as the perpetrator may notice it. Third, most app reviews on the Google Play Store express a desire for these apps to be available offline. The emergency alert option, in particular, would be available in offline mode.

Fourth, because the majority of the users were not native English speakers, some people do not use many apps on their mobile phones because they were uncomfortable with English. Even more extreme was the case of illiteracy of a huge number of people. Fifth, the majority of the apps were working on specific smartphone platforms, and not available for all of them, like Android, Windows, or iPhone. Therefore, it was compulsory for the user to select the contact numbers (friends, family, or other trusted persons) with the same smartphone platforms. Sixth, depending on the situation, the user should be able to hide or unhide the alert. Users should also be able to make contact with law enforcement agencies such as the Punjab police-women safety app, LiveSafe, React Mobile, and Guard My Angel. If users choose SMS alert-based emergency assistance, these apps should be able to analyze and assess their safety. Finally, these apps should offer a support service to assist users who encounter problems while using the app.

### 3.4. Limitations

The main limitation of this study was that certain domestic violence applications were not found in the search due to the keywords. To address this issue, we used relevant terms and keywords from library databases [28]. The study was limited to applications for the two most popular operating systems (iOS and Android) and a Google search [48]. The research did not take into account currently existing domestic violence applications for other operating systems, such as Windows Phone, Blackberry OS, or Symbian. Additionally, this review did not include grey literature or unpublished studies. For several apps, the app release date was not published. As a result, the date of the most recent modification was used as a parameter [48].

In comparison to South Asia, Europe, and north-east America, only a small number of apps were launched and updated in East Asia, Oceania, Europe and Asia, and Africa. This made it difficult to make meaningful regional comparisons to South Asia, Europe, or north-east America. The above-mentioned misconception was due to the online availability of apps, which was often limited to its particular implementation region. For example, apps available exclusively to Indian consumers selectively appear on country-specific versions of app reselling platforms, which can be searched through Indian servers only [23].

Additionally, a possible selection bias resulting from the language restrictions of English, Spanish, and Urdu must be taken into account. The study only included apps in languages that the reviewers could understand, potentially excluding relevant apps in other languages. The language restriction, and region-specific app selection biases, were thought to be the reasons for the small number of apps with origins in East Asia and Europe and Asia. However, this illustrated a widespread drawback in systematic app reviews, which were frequently limited to only one language [49,50,51,52,53,54].

The inclusion of the year of release and update of the app as a qualifying criterion from 2012 through 2022 September result in yet another selection bias. Nonetheless, the current systematic review attempts to examine significant innovations in app development, for which a review of the previous ten years was sufficient, and was consistent with earlier systematic reviews of other mHealth applications [49,50,51,52,53,54]. Furthermore, download statistics were an imprecise measure for actual application use. For example, an app can be uninstalled shortly after it was downloaded, or an app’s privacy policy agreement might be rejected by its user, preventing it from being used [23]. We had limited access to the apps due to download restrictions by the app stores and geographical location. That is why a biased sample was collected. To assess the quality and features of the apps, the data was collected from developers’ descriptions, peer-reviewed research articles, critical reviews in blogs, news articles, and other online sources. As a result, a biased quality analysis was conducted from selective apps that were accessible to us. The review did not use a standardized evaluation tool to assess the quality of the apps.

### 3.5. Open-Ended Challenges

As we have seen, a variety of apps worked to prevent violence. All of the issues were discussed in detail in the preceding section. The following are the major issues we have identified:Apps should be supported in all smartphone platforms (iOS, Android, etc.).Internet connectivity issues: alternative mechanism required.Literacy issue: the majority of the apps are in the English language.Storage issue: if we store the data in a mobile phone’s memory.Privacy is one of the major concerns of domestic violence app users.

In addition, we know that victims use emergency assistance apps in the event of an emergency. The victim was not in a position to send out an alert. There should be a mechanism through which, the app analyzes the situation and generates an auto alert to the law enforcement authorities and preselected emergency contacts. It might be possible if the app collects data from the incident via audio, image, or video recording and uses artificial intelligence models to determine the severity of the situation. An alert is generated if the model determines that the situation is critical. This may be possible but it would present additional challenges such as privacy concerns, storage capacity (both local and cloud), internet connectivity (if cloud-based), and battery consumption.

## 4. Conclusions

Violence against women is a persistent and pervasive problem that is growing at an alarming pace. It might be in the physical and linguistic senses, regardless of the families’ economic and literacy backgrounds [55]. Given the high global incidence of domestic violence [56], this review serves as the first step toward more in-depth research on mobile approaches to addressing domestic violence. This research paper observed at a variety of domestic violence prevention apps from the perspective of their features, benefits, and issues. We present some of the issues that should be considered in future investigations of domestic violence apps.

Emergency assistance, avoidance, informative, self-assessment, and legal information apps were five categories of apps for domestic violence prevention discussed in this systematic review. The majority of apps fall into the first category, which sends out alert messages or makes phone calls to deal with emergencies. Some apps worked to raise awareness about domestic violence or abuse on an individual basis through a questionnaire and the attachment of documents (domestic violence). Victims who do not seek help due to a delay in response time, the need for paperwork, or the fear of being judged by police stations and criminal justice interventions can benefit from mobile apps. Traditional approaches to violence prevention do not provide the same level of security as mobile phone-based violence prevention. The potential purposes, risks, and effects of apps and app functions addressing violence against women must be assessed more thoroughly. Furthermore, issues of safety must be addressed to ensure that women are not put at risk by using mobile technology inadvertently [57]. This was particularly true in developing countries, where mobile phones were frequently shared [58]. To improve functionality and comfort, these apps should be evaluated from the perspective of the users by conducting surveys and collecting reviews.

In our review, we notice that these apps work by having someone generate an alert in the event of an emergency, which is not a real-time approach. The prior and important feature of the potential app for preventing domestic violence could be an auto-alerting system in case of emergency. There is a need for an intelligent model that could examine the current situation in real-time, understand the nuances of the situation, and send an alert to emergency contact numbers in the event of a violent incident. The apps should be more secure and hidden from the perpetrators. It must guide the user regarding domestic violence laws and provide the shelter option when the victim is in an emergency. This paper may also be useful to policymakers who are responsible for developing and enforcing domestic violence laws.

We discussed the best distinctive features of existing domestic violence apps, which will assist the app developers in developing the best app to prevent domestic abuse. Overall, current smartphone technologies offer solutions to domestic violence and they can augment protection provided by traditional methods. Apps currently have many limitations. Incorporating some changes into existing domestic violence prevention apps may improve their functionality and outcome. However, future apps should focus on automation, making better use of artificial intelligence deploying multimedia (voice, video, image capture, text, and sentiment analysis) to aid in live analysis of the situation for accurately generating emergency alerts. Future research may integrate pitch detection and speech recognition, resulting in an extremely powerful and helpful tool. Google speech recognition is a powerful tool which supported multiple human languages. Using Google’s advanced machine learning algorithms and an easy-to-use API can accurately translate speech to text in over 125 languages and variations [59].

Overall, this systematic review on app-based interventions for domestic violence prevention provides important insights for various stakeholders. The study highlights the potential of technology to assist in preventing domestic violence and provides guidance for app developers, researchers, public health practitioners, and policymakers to improve and implement app-based interventions. By taking these findings into account, stakeholders can work together to develop more effective and accessible app-based interventions to prevent domestic violence and support victims.

## Figures and Tables

**Figure 1 ijerph-20-05246-f001:**
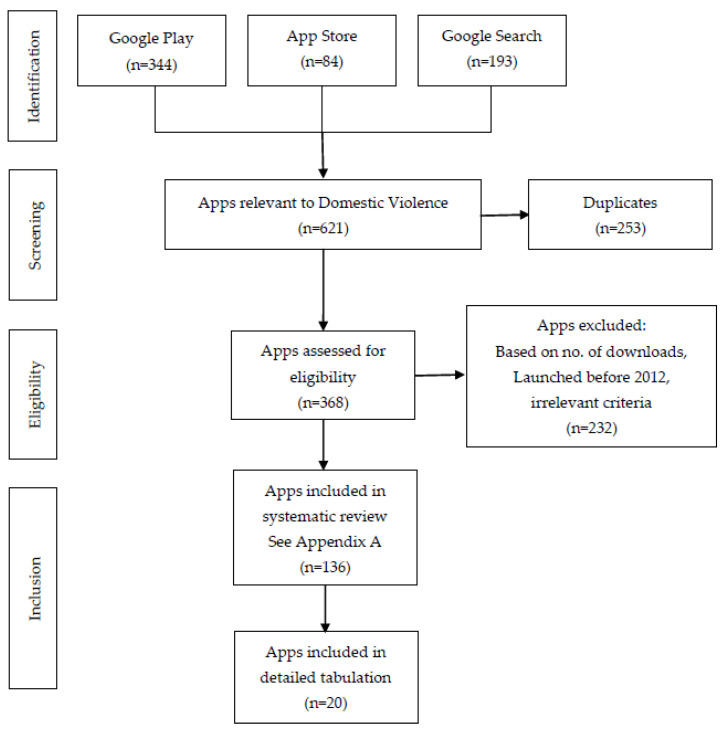
Flow diagram of the systematic review.

**Figure 2 ijerph-20-05246-f002:**
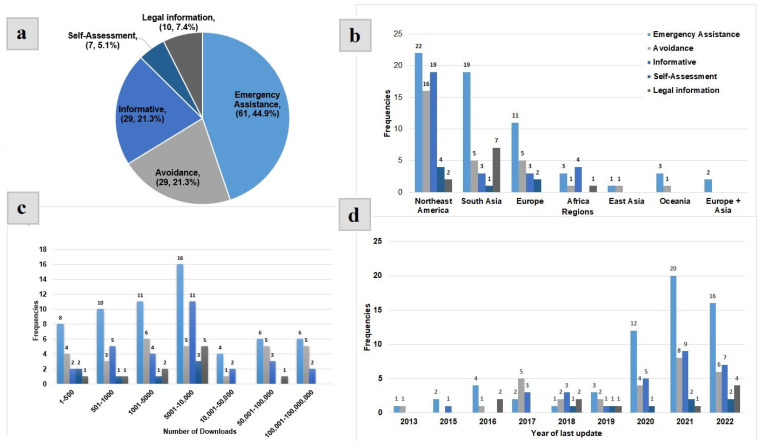
Distribution of 136 apps: (**a**) five functional categories (data presented as number and percentage), (**b**) region, (**c**) number of downloads, and (**d**) year of last update.

**Table 1 ijerph-20-05246-t001:** Detailed features of domestic violence apps.

Category	Features	Distinctiveness	Critical Review
**Emergency Assistance**	Alert local police and security agencies Call to rescue services Location sharing Live chat options Location reviews Security warnings SOS button: user can send immediate notice to the personal contact by sharing location and 10 sec audio Multi language support Family alert Medical alert Accelerometer based trigger Generate alert to three preselected individuals or social service organization by shaking their phone GPS tracking	App provides location review option through which a person can view the reviews against location safety before visiting that place [29]. The app has an option for deaf people to use this app and it can send a warning if the area have any security issues [30]. Complaint system against domestic violence [31]. Accelerometer based trigger is available that automatically set an alert in case the phone detect a fall [32]. App generates an alert by shaking their phones [33].	No auto alert system Not working in offline Mode. No analysis: the applications acquire data from users but do not analyze it in order to make judgments. Uses phone memory for storage Some apps have login system which is not good in emergency conditions.
**Avoidance**	Call 911 or the organizational security SafeWalk option Safety map (i.e., police stations, Hospitals, fire stations etc.) Report tips by sharing suspicious activity by sending picture, audio and video Chatbot: user enters their text, images and videos. Analysis done by using NLP algorithm called the classifier. The classifier then uses confidence levels to determine what labels to show. Circles to create group in the app to communicate with family and friends. App contain documents about awareness of healthy relationships, safety, explaining abuse, safety plans etc. Some of these apps have the “help me” option, which allows the user to send alert at registered emergency contacts and organizational authorities by sending SMS, email with the current location of user. User can post on Twitter and Facebook. App has a “follow me” option, which permits the users, allow their contacts a virtual watch on their activity if user is feeling unsafe.	The apps have option to report a suspicious activity by sharing pictures, audio and video of that location [34]. The app provides geo fences facility [35]. Ref. [36] has access to local hotlines such as, suicide prevention, HIV info, national domestic violence etc. Optional siren and flashlight option available [37]. App offers a support system for traumas and anxiety related responses for reassurance [38]. The battery is low or runs out of power, the app work to tackle the violent situation [39]. User’s medical information file will share in case of emergency [40]. Ref. [41] allows user to call 911 and this call can be canceled within 6 s.	No alert system Not working in offline Mode Majority of the apps have no audio and video facility.
**Informative**	Provide help by hiding information and look like a news and weather app. It has a Go button in the help section to allow the user to send prerecorded audio and written messages to police and trusted person Location map Recording services The triple tap option can be used to generate an alert to a trusted person Interactive map The app provides a list, addresses, and contact number of shelter homes. SafeNight offer victims temporary emergency shelter. The app allows donors to sponsor a night in a hotel for a person in need.	Apps allows the user to send prerecorded voice and text messages to police or the selected person [42]. Interactive map provides a list, addresses, and contact number of shelter homes [43]. App provide the shelter details of the region to offer victims temporary emergency shelter by donating one night shelter rent [44].	Some apps do not have alert system Not working in offline Mode No analysis It covers only small region to provide shelter.
**Legal Information**	No need to download app, this tool can be used on any device, anywhere and any time Image uploads Secure communication by staying anonymous User controlled reporting Helps to pursue legal action against abuser Data collected from the user stored on the device and encrypted	User can anonymously communicate with others. Data stored in encrypted form for the sake of security [45].	Premium app No alert system No video and audio recording No analysis
**Self-Assessment**	Relationship assessment questionnaire Risk analysis via questionnaire User should answer series of question It connects her with nearby emergency shelter or domestic violence hotline using GPS. Hiding feature is available Support services by sharing location Domestic abuse awareness documents, safety guide, dispelling myths (wrong perception about domestic abuse) etc.	The app provide relationship assessment questionnaire and user can call 911 through app [46]. The app provides a checklist of safety behaviors such as, which place is safe to go and it may suggest someone from whom survivor can ask for an emergency loan limited to USD 100 [47]. Some law related documents are available to provide awareness regarding domestic violence.	No alert system Some features of the app are paid.

## Data Availability

The data presented in this paper are available in Appendix A.

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
