# Peer review of "Smartphone Apps for Domestic Violence Prevention: A Systematic Review"

_ijerph, 2023, doi:10.3390/ijerph20075246_

Round 1

Reviewer 1 Report

Excellent review of a much under-researched aspect of domestic violence interventions.  There are a few minor issues that must be addressed to strengthen the manuscript.  

-pg 2 "As previously..." this sentence is out of context and doesn't make sense with the rest of the paragraph

-pg 2 what does "mobile connections" refer to? define term?

-pg 2 reword "treat diseases" to "manage diseases", apps can't treat diseases

-why were some regions selected for review and not others?  That most of the research comes from some regions is apparent but not a solid justification for excluding other regions.  As well, you appear to have included all regions in your review, therefore this statement in the inclusion/exclusion criteria doesn't make sense.  

-What does "no analysis" mean from Table 1?

-The potential of artificial intelligence is mentioned in the abstract, but the potential benefits of this technology is overstated given how much difficulty speech recognition and video recognition of non-white voices and faces still experiences. As the authors noted, the majority of app users are non-English speaking, so this is just not an option for them at this time (or likely in the near future).  Therefore, the abstract should mention the importance of removing pay-wall restrictions from apps or making them not reliant on cellular data as opposed to the artificial intelligence comment. 

Author Response

Manuscript Number: ijerph-2061548

Reviewer’s 1 comments

1

pg 2 "As previously..." this sentence is out of context and doesn't make sense with the rest of the paragraph

We have removed the sentence as advised.

2

pg 2 what does "mobile connections" refer to? define term?

Mobile connections refer to the communications between a mobile device and a cell tower. The connection to the mobile network “Mobile Number” that identifies a person and that gives access to the Service.

We have added a sentence explaining the term mobile connections as requested. The changes appear in the second paragraph of introduction section (page 2) of the revised manuscript as follows:

Changes to the manuscript:

“The communications between a mobile device and a cell tower are referred to as mobile connections [8]”

3

pg 2 reword "treat diseases" to "manage diseases", apps can't treat diseases

Thank you. We have modified the text accordingly.

4

why were some regions selected for review and not others?  That most of the research comes from some regions is apparent but not a solid justification for excluding other regions.  As well, you appear to have included all regions in your review, therefore this statement in the inclusion/exclusion criteria doesn't make sense.

Some regions got excluded due to following reasons:

-download restriction of apps

-the lack of apps available with distinctive features.

-duplicate apps

We have modified the text accordingly in section 2.3 (page 5) of the revised manuscript.

Changes to the manuscript:

“Applications were chosen from certain geographical regions based on the limitations placed by the download availability of apps, the lack of apps available with distinctive features and duplication of apps.”

5

What does "no analysis" mean from Table 1?

No analysis indicates that the applications acquire data from users but do not analyze it in order to make judgments. We suggested to use machine learning on the collected data to make decisions (i.e., auto alert etc.) on the basis of analysis of the domestic violence situation.

The following modification occurs in table 1's critical review column (page 7) of the revised manuscript:

Changes to the manuscript:

“No analysis: the applications acquire data from users but do not analyze it in order to make judgments.”

6

The potential of artificial intelligence is mentioned in the abstract, but the potential benefits of this technology is overstated given how much difficulty speech recognition and video recognition of non-white voices and faces still experiences. As the authors noted, the majority of app users are non-English speaking, so this is just not an option for them at this time (or likely in the near future).  Therefore, the abstract should mention the importance of removing pay-wall restrictions from apps or making them not reliant on cellular data as opposed to the artificial intelligence comment.

Google speech recognition supported multiple human languages. Using Google's advanced machine learning algorithms and an easy-to-use API, accurately translate speech to text in over 125 languages and variations. In our future research, we may integrate pitch detection and speech recognition, resulting in an extremely powerful and helpful tool.

We modified the abstract (page 1) and conclusion (page 10) of the revised manuscript accordingly.

Changes to the manuscript:

“, speech recognition and pitch detection”

“Future research may integrate pitch detection and speech recognition, resulting in an extremely powerful and helpful tool. Google speech recognition is a powerful tool which supported multiple human languages. Using Google's advanced machine learning algorithms and an easy-to-use API, accurately translate speech to text in over 125 languages and variations [58].”

Reviewer 2 Report

This is a timely and important topic, and I appreciate the others for putting it together. The manuscript as well, written, and parsimonious. A few direct questions and comments are as follows:

- if possible, I would like to see you figures in color, considering this is a largely digital publication I don't know that there would be any consideration for increased charges for print, and if so, I would think the publication and editor should consider removing those. Having the figures in color would be particularly helpful in distinguishing one from the other.

- I would also adjust table to slightly in order to distinguish more clearly between the categories. Perhaps an alternating background shading or something would suffice. Whatever you choose I'm sure it would be fine, but I would envision this piece to be more than a review, and summary of applications, and to move forward, also as a resource guide for agencies and organizations to understand more about the market, and what's available both for their current use, and for future development purposes.

- I think a little bit of expansion in the critical analysis section could be helpful. A lot of people are excited about technology, but if technology does not make life easier or safer for people, then it's really just buttons. These applications could be incredibly helpful, but finding ways to make sure people, by-in and can use them safely is incredibly important. Perhaps there's a tiny bit more in the literature or some thing that can add to that critical analysis portion a bit. Particularly the first part in section 3.3.

- love the appendix

- All in all I think this is an incredibly important addition to the literature. Making sure this is available and open access is a high priority. I think the authors of applied systematic review process in a way that is helpful and innovative. My feedback is to accept contingent of a few revisions (as mentioned above).

Author Response

Manuscript Number: ijerph-2061548

Reviewer’s 2 comments

1

if possible, I would like to see you figures in color, considering this is a largely digital publication I don't know that there would be any consideration for increased charges for print, and if so, I would think the publication and editor should consider removing those. Having the figures in color would be particularly helpful in distinguishing one from the other.

We appreciate the suggestion. We have applied the formatting advised on figures.

2

I would also adjust table to slightly in order to distinguish more clearly between the categories. Perhaps an alternating background shading or something would suffice. Whatever you choose I'm sure it would be fine, but I would envision this piece to be more than a review, and summary of applications, and to move forward, also as a resource guide for agencies and organizations to understand more about the market, and what's available both for their current use, and for future development purposes.

We have modified the table accordingly.

3

I think a little bit of expansion in the critical analysis section could be helpful. A lot of people are excited about technology, but if technology does not make life easier or safer for people, then it's really just buttons. These applications could be incredibly helpful, but finding ways to make sure people, by-in and can use them safely is incredibly important. Perhaps there's a tiny bit more in the literature or some thing that can add to that critical analysis portion a bit. Particularly the first part in section 3.3.

We have expanded critical analysis in section 3.3 (page 9) of the revised manuscript as suggested.

Changes to the manuscript:

“The apps, according to Table 1, serve as a central point of contact for victims and rescue resources. These apps are only effective in emergencies since they trigger an alarm. Alarm generation is very simple by clicking one alarm button, shake, or scream. Some apps enable the users by guiding them on how to avoid domestic violence incidents before they happen. When going out alone, some apps, such as StaySafe, BSafe, and My Angel Guard, have a feature called "Follow Me" or "Watch over Me," which allows the user to share their location and status with contacts. These apps are also used to combat street violence or crime.

ShelterSafe is an online database having interactive map used to connect women with the nearest shelter that can provide them with protection, courage, and assistance [41]. A map serves as both a tool and a reminder of the safety net that has been established across Canada to assist victims of domestic violence. It allows the victims to hide the website by redirecting to another website, which encourages the victim to seek help by preventing the abuser from seeing her [41, 42].”

4

love the appendix

Thank you.

5

All in all I think this is an incredibly important addition to the literature. Making sure this is available and open access is a high priority. I think the authors of applied systematic review process in a way that is helpful and innovative. My feedback is to accept contingent of a few revisions (as mentioned above).

Thank you for your time to review our work and for recognizing it as being “helpful and innovative”.
